# Reduction of Thoracic Hyper-Kyphosis Improves Short and Long Term Outcomes in Patients with Chronic Nonspecific Neck Pain: A Randomized Controlled Trial

**DOI:** 10.3390/jcm11206028

**Published:** 2022-10-13

**Authors:** Ibrahim Moustafa Moustafa, Tamer Mohamed Shousha, Lori M. Walton, Veena Raigangar, Deed E. Harrison

**Affiliations:** 1Department of Physiotherapy, College of Health Sciences, University of Sharjah, Sharjah P.O. Box 27272, United Arab Emirates; 2Faculty of Physical Therapy, Cairo University, Giza 12511, Egypt; 3Private Practice and CBP Non-Profit, Inc., Eagle, ID 83616, USA

**Keywords:** neck pain, thoracic kyphosis, randomized trial, postural kyphosis, sensorimotor control

## Abstract

This study investigates thoracic hyper kyphosis (THK) rehabilitation using the Denneroll™ thoracic traction orthosis (DTTO). Eighty participants, with chronic non-specific neck pain (CNSNP) and THK were randomly assigned to the control or intervention group (IG). Both groups received the multimodal program; IG received the DTTO. Outcomes included formetric thoracic kyphotic angle ICT—ITL, neck pain and disability (NDI), head repositioning accuracy (HRA), smooth pursuit neck torsion test (SPNT) and overall stability index (OSI). Measures were assessed at baseline, after 30 treatment sessions over the course of 10 weeks, and 1-year after cessation of treatment. After 10 weeks, the IG improved more in neck pain intensity (*p* < 0.0001) and NDI (*p* < 0.001). No differences were found for SPNT (*p* = 0.48) and left-sided HRA (*p* = 0.3). IG improved greater for OSI (*p* = 0.047) and right sided HRA (*p* = 0.02). Only the IG improved in THK (*p* < 0.001). At 1-year follow-up, a regression back to baseline values for the control group was found for pain and disability such that all outcomes favored improvement in the IG receiving the DTTO; all outcomes (*p* < 0.001). The addition of the DTTO to a multimodal program positively affected CNSNP outcomes at both the short and 1-year follow-up.

## 1. Introduction 

Neck pain is the fourth leading cause for sustaining years of disability with an annual prevalence exceeding 30%, most often in females [1]. Biomechanically, the cervical, thoracic, and lumbar spines are interrelated [2]. Although structural causes of neck pain are not completely understood, they are believed to be related to the interrelated functions of anatomical structures connected to the cervical spine [3]. Potentially, any event leading to altered joint mechanics or muscle functions can cause neck pain [4]. 

The thoracic spine acts as a base of support for the cervical spine and influences its kinematics through the cervicothoracic junction [3]. Several studies have highlighted the effect of thoracic spine abnormalities on the kinematics of the cervical spine [5,6,7]. Specifically, mobility restrictions in the cervico-thoracic and upper thoracic regions were reported to be associated with neck pain [5,6]. Furthermore, it has been reported that the incidence of neck disorders is increased in older adults with a concomitant higher prevalence of thoracic hyper-kyphosis [6]. This would implicate postural impairments in the thoracic spine leading to a dysfunction of cervico-thoracic musculature such as serratus anterior, levator scapulae, and trapezius [8,9,10]. 

Lau et al. reported a positive correlation between a higher upper thoracic angle and neck pain, but failed to link this to neck pain intensity [3]. In addition, Kaya and Çelenay reported a positive correlation between thoracic curvature and neck pain and reported a negative correlation with neck pain intensity [11]. Furthermore, neck pain populations have been reported to have reduced trunk rotations during different speeds of walking [12]. 

Because changes in sagittal thoracic alignment have been reported to alter the mechanical loading of the cervical spine [10,13] and decreased thoracic mobility has been identified as one of the predictors for neck and shoulder pain [3], it makes sense that thoracic articular treatment improves local kinematics and that simultaneously neck pain improves [7,8,14]. 

Thoracic kyphosis has not been uniformly correlated with neck pain intensity and there is a general lack of investigations determining the role that rehabilitation of thoracic kyphosis plays in improving chronic cervical spine disorders. The purpose of this study was to investigate the immediate and 1-year effects of a multimodal program, with thoracic hyper kyphosis rehabilitation using the Denneroll™ thoracic traction orthosis (DTTO), applied to participants with chronic non-specific neck pain and thoracic hyper-kyphosis. Regarding the DTTO, it is likely that a significant reduction in thoracic kyphosis will occur due to the visco-elastic effect of the three-point bending extension traction during sustained supine loading while on the DTTO; this has been previously reported for extension traction devices for all regions of the sagittal plane of the spine [15,16,17].

The study hypothesis is two-fold: (1) the DTTO, as a three-point bending thoracic extension device, will cause a significant reduction in thoracic kyphosis; and (2) that the reduction in thoracic kyphosis will improve the short and long-term outcomes of participants with chronic non-specific neck pain.

## 2. Methods

A prospective, investigator-blinded, parallel-group, pilot randomized clinical trial was conducted at a research laboratory of our university and was retrospectively registered with the Pan African Clinical Trial Registry (PACTR2019107484227). Recruitment began after approval was obtained from the Ethics Committee of the Faculty of Physical Therapy, Cairo University with the ethical approval No. Cairo -6-2018-11M.S. Following Ethics Committee approval, participant recruitment began in September 2018. The participants were followed up for 1 year (till 2019 October); all participants signed informed consent prior to data collection. The reason behind the retrospective registration was that legislation in Egypt only requires local registration for clinical trials and this what was completed at the outset by prospectively registering in a non-WHO-approved registry.

We recruited a sample of 80 patients from our outpatient facility at the University of Cairo. The Consort participant flow diagram for our study is shown in Figure 1. Participants were screened prior to inclusion by measuring the sagittal thoracic kyphotic angle ICT-ITL (max) using a 4D formetric device (Figure 2). After being screened by a physiotherapist, all potential participants were invited to undergo comprehensive assessment by an orthopedist where other causes of thoracic kyphosis were excluded. Participants were included if the angle measured more 55 degrees. Furthermore, the patients were included if they had chronic nonspecific NP lasting for at least 3 months, and were able to read and speak English.

Exclusion criteria included the presence of any signs or symptoms of medical “red flags”, a history of previous spine surgery, signs or symptoms of upper motor neuron disease, vestibular basilar insufficiency, amyotrophic lateral sclerosis, bilateral upper extremity radicular symptoms, a history of spinal column fracture, spinal tumors and related malignancies, congenital spinal anomalies, cancer, or rheumatoid arthritis. Furthermore, individuals with spinal scoliosis were excluded.

Participants were randomly assigned to an intervention group (*n* = 40) or control group (*n* = 40) according to a random number generator and restricted to permuted blocks of different sizes, with the researcher blinded to the sequence designated for each person. The participants in both groups completed a 10-week, 3× per week, 30 sessions total multimodal program consisting of physical pain relief methods, thoracic spine manipulation, myofascial release, and therapeutic exercises. The beneficial effects for this multimodal program have been previously reported [1,14,18,19,20].

### 2.1. Multimodal Program

The multimodal program was delivered by the same physiotherapist, with 10 years of experience and training in the specific manual techniques in order to minimize inter-therapist variation and enhance fidelity. The participants in both the control and intervention groups received the multimodal program. Both groups received the same length of multimodal treatments and the sessions lasted 30–45 min each. However, the participants in the intervention group received an extra intervention (and time) using the Denneroll™ traction device. Thus, we attempted to provide the same time of attention equivalence in each group provided by the treating therapist during the intervention sessions.

### 2.2. TENS and Heat Therapy

The participants in both groups received conventional TENS therapy (20 min). TENS was applied over the painful area, using a frequency of 80 Hz; pulse width of 50 µs; intensity (mA) set at the person’s sensorial threshold; modulation up to 50% of variation frequency; symmetrical, and rectangular biphasic waveform. These parameters were set for an optimum analgesic effect [18]. Moist hot packs (15 min) were applied prior to electrical stimulation. The TENS and heat therapy were repeated three times per week for 10 weeks.

### 2.3. Soft Tissue Mobilization

Soft tissue mobilization was performed on the muscles of the upper quarter with the involved upper extremity positioned in abduction and external rotation to preload the neural structures of the upper limb [19]. Manual pressure was applied to the soft tissues of the upper quadrant in a deep, stroking manner. The therapist concentrated on any tissues on the cervical and scapular region and upper extremity that were graded as tight or tender in the evaluation. This soft tissue mobilization was repeated three times per week for 10 weeks (30 sessions, 20 min face-to-face sessions)

### 2.4. Thoracic Spine Manipulation

Following the protocol previously outlined by Flynn [21], the participants were placed in the supine position with their arms crossed and with one hand, the clinician established a hand contact over the inferior vertebra of the identified hypomobile motion segment. With their second hand, a downward high velocity thrust was applied with the weight of the clinician’s body through the patient’s elbows or forearms. This procedure was performed at each identified segment with extension restriction range of motion determined clinically.

The initial treatment for all patients included thrust manipulation procedures consisting of a high-velocity, low amplitude end-range procedure, directed at the upper, mid, and lower spines of spinal segments identified as hypomobile during segmental mobility testing. Therapists were required to perform at least 1 technique targeting the upper thoracic spine, 1 technique targeting the mid thoracic spine, and 1 technique targeting the lower thoracic spine during each visit for each patient. If a pop (cavitation) occurred, then the therapist moved on to the next procedure. If not, the participant was repositioned, and the technique was performed again. This procedure was performed for a maximum of 2 attempts.

### 2.5. Functional Exercises

A functional and strengthening exercise program was administered that focused on deep cervical flexors, shoulder retractors, and serratus anterior activation and was conducted according to the protocol described in Harman et al. [20].

Strengthening deep cervical flexors through chin tucks in supine lying with the head in contact with the floor, the progression of this exercise was to lift the head off the floor in a tucked position and hold it for varying lengths of time (this was to progress by two second holds starting at two second i.e., 2, 4, 6, and 8 s). Shoulder retractors were strengthened first while standing using a TheraBand™ by pulling the shoulders back; then the participant was progressed to shoulder retraction in the prone position using weights. In the standing position, the patient was asked to pinch their scapulae together without elevation or extension in the shoulder holding this position for at least six second then relaxing. Participants performed each of these progressive exercises for two weeks prior to advancing to a more difficult version. At the consultation, if they could complete 3 sets of 12 repetitions correctly for the strengthening, they were progressed to the next exercise.

The progression of exercises was as follows:(1)TheraBand™;(2)3 lbs;(3)3 lbs and TheraBand™;(4)5 lbs;(5)5 lbs and TheraBand™;(6)8 lbs;(7)Using 8 lbs and TheraBand™.

The dynamic hug was performed to strengthen the serratus anterior while standing with the back toward the wall. The participant began with the elbow flexed 45°, the arm abducted 60°, and the shoulder internally rotated. The participant then horizontally flexed the humerus by following an arc described by his hands. Once the participant’s hands touched together, they slowly returned to the starting position. Participants were instructed to complete three sets of 12 repetitions of the dynamic hug exercises. The complete functional exercise program was to be repeated three times per week for 10 weeks.

The participants in both groups were instructed to perform neck retraction/extension, scapular retraction, and deep upper cervical flexor strengthening exercises at home, twice daily as their home routine. To monitor the exercise frequency performed during the study, participants were given a pamphlet illustrating the exercises and a record sheet and were instructed to record the time and sets of the home exercises. Mean exercise frequency per week and mean exercise duration per day were recorded. Participants were encouraged to perform all exercises at least twice a week for up to one year after treatment. All persons were contacted by telephone every three months to collect the record sheets and encouraged to maintain the training.

### 2.6. Denneroll™ Thoracic Traction Orthotic (DTTO)

In addition, the participants in the intervention group received the DTTO (Denneroll Industries, Sydney, NSW, Australia), solely during the clinical setting. Thus, the only difference in treatments between the intervention and the control group was the application of the DTTO Figure 3. The participants were instructed to lie flat on their back on the ground with their knees slightly bent at 20–30° for comfort and arms gently folded across their stomach. The examiner positioned the apex of the DTTO in one of three regions: lower thoracic (T9–T12); mid-thoracic (T5–T8); and upper-thoracic (T1–T4) depending on the apex of each participant’s thoracic kyphosis deformity. For lower thoracic kyphosis (T9–T12) the DTTO is turned 180° so the peak contacts the lower thoracic spine (T10) while the tapered end supports the mid thoracic region (Figure 3). For persons with mild–moderate posterior thoracic or backwards tilt translation postures with more of an upper thoracic kyphosis and anterior head translation, the DTTO is placed centered on top of a 20 mm block in order to cause anterior shift of the thoracic spine; set up not shown. All participants began at 3-min per session of DTTO application; at each visit they were encouraged to increase the duration by 2–3 min, until such time they were able to reach the goal of 15–20 min per session.

### 2.7. Outcome Measures

A series of outcome measures were obtained at three intervals: (1) baseline; (2) one day following the completion of 30 visits after 10 weeks of treatment; and (3) one year after the participants’ 30 session re-evaluation. The sequence of measurements was identical for all persons. Outcome measures included: (1) kyphotic angle ICT-ITL (max) as a primary outcome; (2) neck pain and disability (NDI); (3) sensorimotor control outcomes; (4) head repositioning accuracy (HRA); (5) smooth pursuit neck torsion test (SPNT); and (6) overall stability index (OSI) as secondary outcomes. All outcome assessments were carried out with two data collectors who were blinded to group allocation to prevent potential recorder and ascertainment bias. Participants were blinded to their measurement scores to address potential expectation bias and were instructed not to inform the assessors of their intervention status.

### 2.8. ICT-ITL (Max)

Thoracic kyphosis was assessed using a valid and reliable [22], 4D formetric device where determination of thoracic kyphosis angle ICT-ITL (max) is measured between tangents from the cervicothoracic junction (ICT-T1) and that of the thoracolumbar junction (ITL-T12). Participants were included if the angle measured 55° or more [23]. There was a good correlation between the formetric vs. Cobb angle of thoracic kyphosis (Pearson’s r correlation = 0.799) but formetric measurements consistently over-estimate thoracic kyphosis by an average of 7° [23]; indicating that the T1–T12 radiographic would be a minimum of 48°, which is the upper end of normal in young adults, when the formetric angle is 55° [24]. See Figure 2 and Figure 4 for the formetric analysis in our participants.

### 2.9. Neck Disability Index

The neck disability index (NDI) to assess activities of daily living impact was administered. The NDI has good reliability, validity, and responsiveness to change [25].

### 2.10. Numerical Rating Score (NRS)

Neck pain average intensity over the previous week was assessed using a 0–10 NRS where 0 = no pain, …, 10 = bed ridden and incapacitated. The reliability [26] and validity [27] of the NRS is good to high.

### 2.11. Sensorimotor Control Measures

Assessment of sensorimotor function included: (1) cervical joint position sense testing; (2) head and eye movement control; and (3) evaluation of postural stability.

### 2.12. Cervical Joint Position Sense Testing

The valid and reliable technique [28] of head repositioning accuracy (HRA) assessment with the CROM device was performed according to a previous protocol [29]. In an upright seated posture on a stool with no backrest, the CROM device was placed on the participant’s head, both feet were firmly on the floor with knees bent at an approximate 90° angle. The neutral head position (NHP) was established as the beginning and reference positions where the CROM device was adjusted to zero for the primary plane of rotational movement. Individuals were instructed to close their eyes, memorize the starting position, actively rotate their head 30° about the vertical axis, and reposition their head to the starting position with no requirements for speed, only accuracy was encouraged. HRA was measured as the difference in degrees in the primary plane of movement between the origin and the return positions [30]. Participants performed three repetitions within 60 sec in each rotational left and right directions, for a total of six sessions.

### 2.13. Head and Eye Movement Control: Smooth Pursuit Neck Torsion Test (SPNT)

Electro-oculography was used for the SPNT, which is an accurate means of assessing disturbances in eye movement control [31]. The method has been described elsewhere in detail [32]. The test was performed with the participant’s head and trunk in a neutral forward position and then a trunk rotation position (head neutral, trunk in 45° rotation). The participants were instructed to perform three blinks (for recognition and elimination in data analysis) and then to follow the path of a light as closely as possible with their eyes. The SPNT test value was calculated as the difference between the average gain in the neutral and torsion positions for both left vs. right rotation.

### 2.14. Postural Stability

Postural stability was evaluated with a Biodex Balance System SD (BBS) (Biodex Medical Systems, Inc., Shirley, NY, USA). Dynamic balance testing was assessed allowing simultaneous displacements in both the anterior/posterior (AP) and medial/lateral (ML) directions. BBS measures the deviation of each axis in the horizontal plane of the platform during dynamic balance assessments and reports indices for ML, AP, and an overall stability index (OSI) whereby a reduced balance correlates with large variance. Balance indices were calculated over three 10-s trials, with 20 s of rest between trials; the average of the three trials was recorded. The BBS was set to a dynamic position of 4 out of 8 [33]. Several studies have used the device and have been proven to be reliable and valid for clinical studies [34,35,36].

All outcome assessments were carried out by 2 assessors blinded to group allocation. The: kyphotic angle ICT-ITL (max); neck pain and disability (NDI); sensorimotor control outcomes; head repositioning accuracy (HRA); and overall stability index (OSI) were performed by a physiotherapist with 20 years of experience in these measurement techniques (T.S). The SPNT was conducted by an ophthalmologist (not an author) with 5 years of experience (R. W., MD).

## 3. Statistical Analysis

### 3.1. Sample Size

A priori sample size calculation based on a non-published pilot study conducted for 9 patients, indicated that 35 participants per each group were required to detect an effect size of 0.7 at 80% power and a significance level of 0.05 (5% chance of type 1 error). The mean difference of the primary outcome thoracic kyphosis angle ICT-ITL (max) was 11 and the standard deviation of this differences was 15. To account for possible drop-outs, the sample size was increased by 10% to 40 per group.

### 3.2. Data Analysis

Variance homogeneity was tested with Levene’s test, obtaining a 95% confidence level and *p*-value > 0.05, and confirming variance equality. Descriptive statistics (means  ±  SD unless otherwise stated) were summarized at each time point. Student’s *t*-test for continuous variables or chi-squared for categorical variables were performed.

The design used an intention-to-treat approach with alpha set at 0.05 level of significance for all analyses. Comparative treatment effects of the two alternative treatments over the course of the 1-year follow-up were examined with two-way analysis of covariance with repeated measures, followed by the Bonferroni post hoc test. The models included one independent factor (group), one repeated measure (time), and an interaction factor (group × time) and gender as covariate If interactions were found (*p* < 0.05), the baseline value of the outcome as covariates was used to assess between group differences. Cohen’s d was calculated to examine the average impact of the intervention [37].

All data were analyzed using SPSS version 20.0 software (SPSS Inc., Chicago, IL, USA) with normality and equal variance assumptions ensured prior to the analysis.

### 3.3. Imputation of Missing Values

To impute any missing values for the intervention and control groups, we constructed models that included the variables related to the missing data and the variables correlated with that outcome. The main cause of the missing data was patient dropout at the long-term follow-up measurement interval at 1 year. The outcome measures at 1-year follow up were missing for three patients from the experimental group and seven patients from the control group (reasons for dropout are depicted in Figure 1). As the missing data were at the end of the trial, the last present value was carried forward. Imputation models included corresponding outcome values measured at baseline, then at 10 weeks. Other variables included in the imputation model were selected based on maximizing the correlation with the variable imputed. The characteristics which were associated with the variable imputed in the regression analysis were age, sex, and smoking status. This imputation created five complete datasets according to Rubin’s method [38]. Pooled results were used for data analysis. We conducted a sensitivity analysis comparing the results from the imputed data to the original dataset, and the results were similar.

## 4. Results

Two hundred participants were initially recruited and screened, of whom 80 met the inclusion criteria and agreed to participate in the study. Three persons in the intervention group and seven in the control group resigned at 1-year follow-up for business and personal reasons. Figure 1 presents this information.

### 4.1. Baseline Demographics and Characteristics

The intervention and control groups were comparable for age, weight, sex, marital status, pain duration, and smoking status, indicating randomization was successful for these variables. Table 1 reports this data.

### 4.2. Between Group Analysis

A general linear model using repeated measurements identified significant group × time effects in favor of the intervention DTTO group for the following outcomes: thoracic kyphosis angle (ICT-ITL (max); NDI, NRS pain intensity; HRA for right and left rotation repositioning accuracy; SPENT, posture stability measured as the OSI. Table 2 reports the thoracic kyphosis outcomes, Table 3 reports the NDI and pain intensity while Table 4 reports the sensorimotor control outcomes.

### 4.3. The 10-Week Evaluation

-Thoracic kyphotic angle

Significant differences were found between groups, favoring the intervention group for kyphotic angle ICT-ITL (max) (*p* < 0.001) with an approximate 19° reduction in kyphosis angle for the DTTO group. These data, including effect sizes for both groups, are reported in Table 2. See also Figure 4 for a representative example of the changes.

-NDI and Pain Intensity

Following 30 treatment sessions, the between-group statistical analysis, showed better improvements for the intervention vs. control group in NDI (*p* < 0.001) and pain intensity (*p* < 0.001). These data, including effect sizes for both groups, are reported in Table 3.

-Sensori-motor control

Both groups improved similarly for two sensori-motor control outcomes where no group differences were found for: left sided HRA (*p* = 0.3) and SPNT (*p* = 0.48). In contrast, the intervention group had significantly greater improvements for two sensori-motor control measurements: right sided HRA (*p* = 0.02) and OSI (*p* = 0.047). These data, including effect sizes for both groups, are reported in Table 4.

### 4.4. One-Year Follow-up

Between group analysis identified a regression back to baseline values for the control group outcomes. Thus, all variables were significantly different favoring the intervention group at 1-year follow-up. Kyphotic angle ICT-ITL (max) maintained its improvement (*p* ˂ 0.001), with an 18° overall improvement from baseline in the DTTO group; see Table 2. Pain and disability were significantly improved in the intervention group vs. the control group: NDI (*p* ˂ 0.001); neck pain intensity (*p* ˂ 0.001). Sensori-motor measures were also significantly improved in the intervention group compared to the control: HRA-right (*p* ˂ 0.001); HRA-left (*p* ˂ 0.001); SPNT (*p* ˂ 0.001); OSI (*p* ˂ 0.001); see Table 3 and Table 4. Cohen’s d and effect size (r) for both groups for all variables are reported in Table 2, Table 3 and Table 4.

## 5. Discussion

The current study presented a two-fold hypothesis: first, that the DTTO would cause a significant reduction in thoracic kyphosis, and two, that the reduction in thoracic kyphosis would improve the short and long-term outcomes of participants with chronic non-specific neck pain with concomitant hyper thoracic kyphosis. The differences between our intervention and control groups identified an 18–19° reduction in thoracic kyphosis in the group receiving the DTTO at both the 10-week and 1-year follow-up, while the control group’s kyphosis angle remained unchanged. Concerning the sensorimotor control group’s measurements at 10-weeks, two out of the four assessments identified a significant difference in favor of the DTTO group (OSI-balance and Right HRA) and at the 1-year follow-up all of the measures were significantly different in favor of the DTTO group. Thus, both of the hypotheses of our investigation were confirmed by these findings. To our knowledge, this is the first study to provide clear evidence that rehabilitation of thoracic hyper-kyphosis influences these specific outcomes in chronic neck pain sufferers with hyper-kyphosis.

### 5.1. Thoracic Kyphosis Improvement

Thoracic hyper-kyphosis represents one of the top four spine abnormalities associated with adult spine deformity (ASD), a world-wide, known set of disabilities affecting adults over the age of 18 years [39,40,41]. For example, Pellise et al. [39]. identified that patients with thoracic kyphosis over 60° had significantly lower health-related quality of life scores compared to patients afflicted with four other major health disorders (Type II diabetes, rheumatoid arthritis, heart disease, pulmonary disease). While 60° is the recommended cut-point for thoracic hyper-kyphosis in ASD populations, other investigations have identified that the cut-point between those with pain, lower self-image, and decreased function is 45° [42,43,44].

Due to the volume of investigations, identifying thoracic kyphosis is a considerable cause of pain, disability, and reduced quality of life outcomes, conservative treatment strategies to reduce its magnitude are critically necessary. To this end, it is generally considered that effective interventions for postural thoracic hyper-kyphosis should include specific rehabilitation exercises and practiced forced idealized posture alignment in stance and in sitting [44,45,46]. In more severe cases, or in cases with Scheuermann’s kyphosis, a sagittal plane corrective orthosis brace is recommended [45].

A recent systematic literature review with meta-analysis identified that strengthening exercises have a considerable effect on thoracic kyphosis reduction when applied over the course of an average of 12.5 weeks with three sessions per week [44]. Considering only the homogenous exercise studies, an approximate reduction in thoracic kyphosis of 5° or less was identified [44]. More recently, in a small scale RCT with low power, Bezalel et al. [46]. identified a significant reduction in thoracic kyphosis (9°–10° reduction) in patients with Scheuermann’s kyphosis receiving the Schroth series of exercises and stretches to reduce kyphosis. Initially, patients had a 60° kyphosis on X-ray (Cobb T3–T10) and inclinometry (T1–T12) that was reduced to approximately 50°.

In the current investigation, we used a four-D formetric scanner to evaluate thoracic kyphosis and our average participant’s kyphosis was 82° which was reduced by 18° down to 64° in the group receiving the DTTO. For comparison, it is known that the formetric and inclinometry measures of external thoracic kyphosis overestimate the radiographic determined thoracic kyphosis by approximately 7° and maybe more depending on the unique population [23,46,47]. It is likely that our current participant population had a radiographic determined thoracic kyphosis that averaged at least 60° depending on the vertebral levels of measurement. Further, we estimated our radiographic kyphosis reduction to be between 12°–15° based on existing comparative population data; making our results one of the largest conservative reductions in thoracic kyphosis reported in an RCT in the literature to date [44,46].

Arguably, adults with a large increased thoracic kyphosis (60°–80°) that is ‘fixed’ (Scheuermann’s kyphosis and other deformities) would seem not to be amenable to physical maneuvers (exercise and manipulation); however, they are able to be reduced with three-point bending thoraco-lumbar braces [45]. Similarly, extension traction devices such as the DTTO use the principles of three-point bending as in braces; although extension traction devices are shorter duration applications with higher loading [15,16,17]. Though we did not specifically investigate the difference between more rigid vs. more flexible thoracic kyphotic deformities, our population did indeed have a large increase in thoracic kyphosis compared to that found in a healthy population [24,42,43,44]. We speculate that the large and significant reduction in thoracic kyphosis found in our DTTO group is due to the visco-elastic effect of three-point bending extension traction during sustained supine loading while on the DTTO. Our results are generally consistent with previous investigations looking at patients treated with different types of thoracic spine three-point bending extension traction devices; however, these previous investigations suffer from a lack of controls and small sample sizes [15]. Future investigations should use radiography to determine the type of thoracic hyper-kyphosis, its flexibility, and its amenability to three-point bending extension devices such as the DTTO.

### 5.2. Pain, Disability, and Sensorimotor Control

The assumption that restoring thoracic sagittal plane posture should improve cervical spine pain and kinematics has evidence in the literature. For instance, it has been proposed that upper thoracic kyphosis increases the T1-slope into a more flexed posture and this, in turn, creates a situation of forward head posture, increased strain on the cervical-thoracic muscles and ligaments [15,39,40]. For example, Kaya and Çelenay reported a positive correlation between thoracic curvature and neck pain [11]. Furthermore, abnormal head posture can result in altered joint position and dysfunction that can lead to pain and abnormal afferent information [10,48].

Forward head translation causes both a reduced range of movement and an altered segmental cervical spine kinematic pattern [10]. Thus, altered sagittal cervical spine alignment from thoracic hyper-kyphosis could potentially result in abnormal sensorimotor integration through changes in afferent input as a direct consequence of altered cervical spine kinematics and altered soft tissue strains [48]. The current study’s findings of reduced neck pain, disability, and improved sensorimotor control in the DTTO group add credence to the above biomechanical and clinical investigations detailing the effects of thoracic spine abnormalities on the cervical spine. Treating the spine as a synchronized kinetic chain should be considered the standard particularly in cases of chronic non-specific neck pain with concomitant thoracic hyper-kyphosis.

### 5.3. Limitations and Summary

The current study has limitations to consider which should lead to future investigations. First, we did not use participant and treatment provider blinding. However, examiners did not discuss the clinical importance of correcting the thoracic kyphosis in either group in order to account for the placebo effect in the DTTO group and a possible nocebo effect in the control group at long term follow-up. Second, the participants were a convenience sample of young adults from an out-patient facility and thus may not be representative of all patients with chronic non-specific cervical spine complaints. Third, the outcome measures we used to verify if correction of thoracic kyphosis alignment improves sensori-motor control, pain, and disability may not be the only or the ideal assessments for CNSNP outcomes. Fourth, we measured the thoracic kyphosis using an external posture assessment device and this does not provide the same quantitative data as radiographic or other advanced imaging methods for measurement of thoracic kyphosis.

Finally, both groups received the same time and number of sessions for the multimodal treatments. However, the participants in the intervention group received an extra intervention (and time) using the Denneroll™ thoracic extension traction device. We attempted to provide the same time of attention equivalence in each group provided by the treating therapist during the intervention sessions. However, as attention and interpersonal interactions alone may influence pain, and other health outcomes, this is a limitation to the study design in as much as the groups did not receive equal interventions. Importantly though, previously, it has been identified that when a placebo device is added to the control groups’ interventions to mimic the time and number of sessions on the Denneroll™ in the cervical spine, that the placebo device did not influence the outcomes of neck pain and disability [49]. Still, this is something that should be addressed in future projects.

## 6. Conclusions

Notwithstanding the study limitations, the unique contribution of the current investigation is that we determined thoracic hyper-kyphosis reduction plays a significant role in improving both the short and long-term outcomes in patients suffering from chronic nonspecific neck pain. In these relevant populations, it would seem of value to rehabilitate thoracic hyper-kyphosis abnormalities towards normal alignment as a primary management strategy. The DTTO investigated in this study is a simple orthotic that can be prescribed for home use or utilized under the supervision of a treating clinician as used in this investigation.

## Figures and Tables

**Figure 1 jcm-11-06028-f001:**
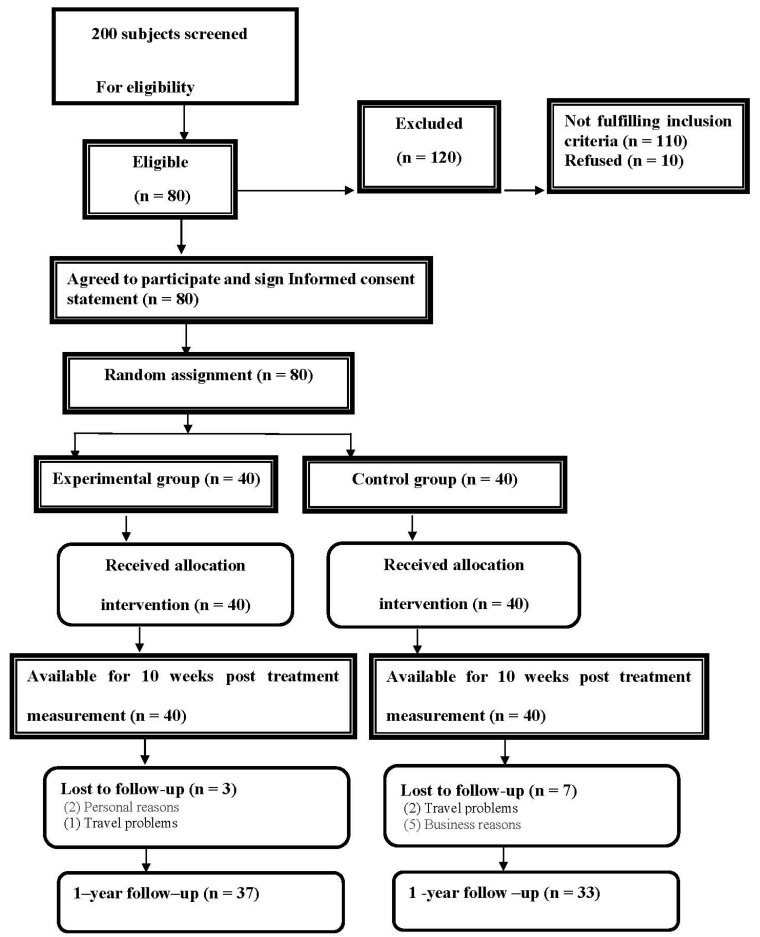
Flow chart of participants in the study over time.

**Figure 2 jcm-11-06028-f002:**
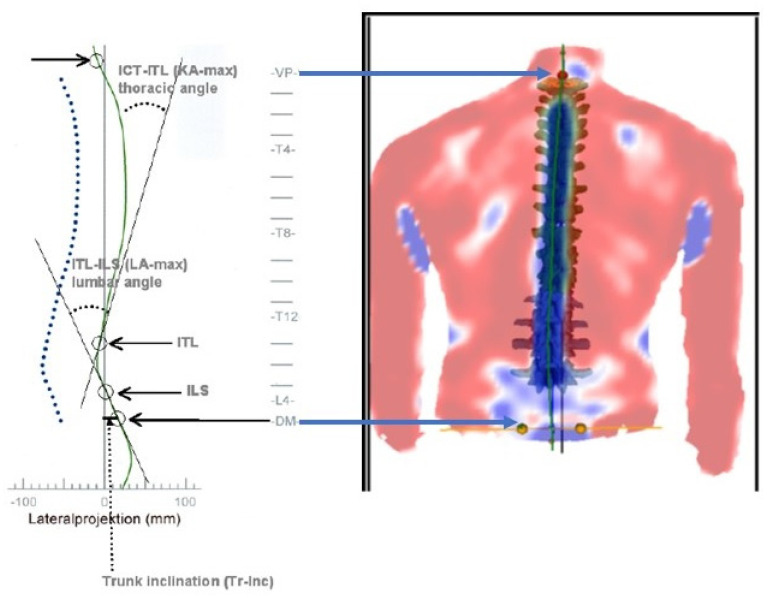
The 4D Formetric device measurement of Thoracic Kyphosis and Trunk Inclination where kyphotic angle ICT-ITL (max) is measured between tangents of cervicothoracic junction (ICT) and of thoracolumbar junction (ITL). ICT: Inflectional points from cervical to thoracic spine. ITL: Inflectional points from thoracic to lumbar spine. KA: kyphosis angle. LA: lordosis angle. VP: Vertebra prominence. DM: Dimple.

**Figure 3 jcm-11-06028-f003:**
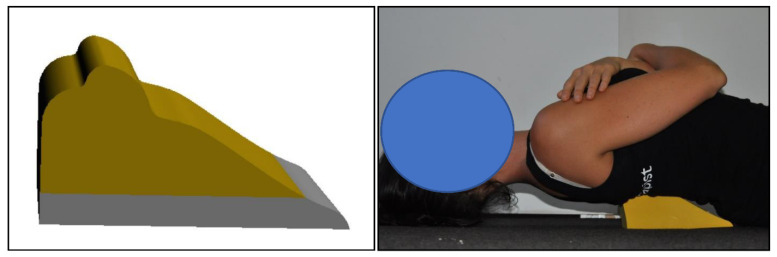
Denneroll™ Thoracic Traction Orthosis (DTTO). The DTTO can be placed in the upper (T3–T4), mid thoracic spine (T5–T8)-shown in B; or lower thoracic region (T9–T12) pending the apex of a participant’s thoracic kyphosis and sagittal balance alignment. Each participant began lying supine over the apex of the DTTO for 1–3 min and progress 1–3 min per session until the target of 15–20 min per session was reached. Images copyright CBP Seminars, Inc. Reprinted with permission. Note: The individual used in the figures in this manuscript was a paid model and provided consent for commercial use.

**Figure 4 jcm-11-06028-f004:**
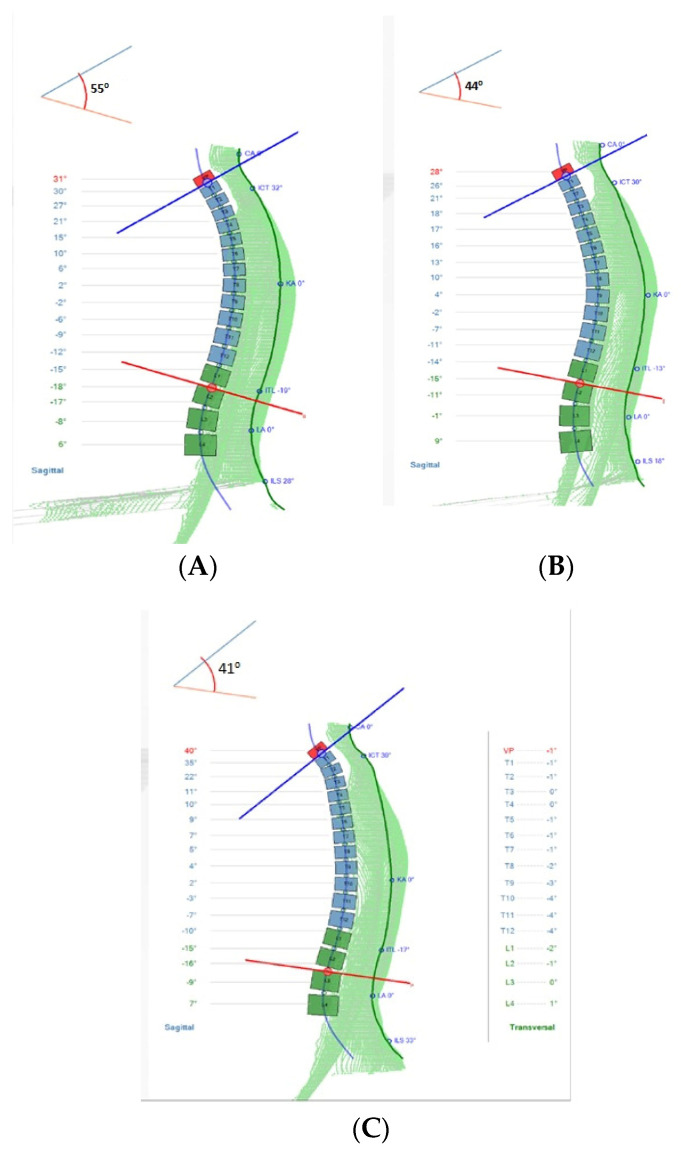
Kyphosis formetric posture alignment outcomes for a sample intervention group participant receiving the DTTO. (**A**) Initial baseline; (**B**) after 10-weeks and 30-sessions of intervention; and (**C**) the 1-year follow-up assessment where no further treatment was provided over the course of one year.

**Table 1 jcm-11-06028-t001:** Baseline participant demographics. Interventional group (Int.) is the group receiving standard care plus the Denneroll™ thoracic traction orthotic (DTTO). Control group (Con.) is the group receiving standard care only. Values are expressed as means ± standard deviation where indicated.

	Int. Group (*n* = 40)	Con. Group (*n* = 40)
Age (y)	25.05 ± 3	24 ± 4.2
Weight (kg)	66 ± 10	60 ± 9
Sex		
Male	28 (70%)	30 (75%)
Female	12 (30%)	10 (25%)
Single	31 (77.5%)	29 (72.5%)
Married	9 (22.5%)	11 (27.5%)
Separated, divorced, or widowed	0	0
Pain duration (%) [Mean ± SD]		
1–3 y	11 (27.5%) [5.3 ± 2]	9 (22.5%) [5.8 ± 1]
3–5 y	16 (40%) [4.9 ± 1.5]	18 (45%) [5.4 ± 1.3]
>5 y	13 (32.5%) [4.8 ± 2]	15 (37.5%) [5.7 ± 0.9]
Smoking		
Light smoker	15 (37.5%)	18 (45%)
Heavy smoker	4 (10%)	2 (5%)
No Smoker	21 (52.5%)	20 (50%)

**Table 2 jcm-11-06028-t002:** The changes in sagittal alignment management outcomes in experimental and control groups vs. time. Kyphotic angle ICT-ITL max = angle of kyphosis between tangents of cervicothoracic junction (ICT) and of thoracolumbar junction (ITL). Values are mean ± standard deviation. G = group; T = time; I = intervention group; C = control group; C.I. [] = 95% confidence interval; *p* = statistical significance; C.I. [] = 95% confidence interval; Cohen’s d value = d; * indicates statistically significant difference.

		Baseline	10-Weeks	1-Year Follow-up	Cohen’s d 10-Weeks vs. Baseline	Cohen’s d 1-Year vs. Baseline	*p*-Value		
							G	T	G vs. T
**ICT-ITL max**	I	82.15 ± 5.3	63.40 ± 6.2	64.6 ± 5.7	d = 3.2	d = 3.18	<0.001 *	<0.001 *	<0.001 *
	C	83.15 ± 4.9	82.2 ± 4.5	83.8 ± 3.8	d = 0.2	d = −0.14			
***p*-value** **C.I.**	0.5 [−4.3, 2.3]		<0.001 * [−22.9, −15.8]	<0.001 * [−22.3, −16.1]					

**Table 3 jcm-11-06028-t003:** The changes in pain and disability outcomes in interventional (DTTO) and control groups vs. time. NDI = neck disability index; Pain intensity is 0–10 where 0 is no pain and 10 is incapacitated; I = interventional group; C = control group; G = group; T = time; G vs. T = group vs. time; all values are expressed as means ± standard deviation; C.I. [] = 95% confidence interval; Cohen’s d value = d; * indicates statistically significant difference.

		Baseline	10-Weeks	1-Year Follow up	Cohen’s d 10-Weeks vs. Baseline	Cohen’s d 1-Year vs. Baseline	*p*-Value		
							G	T	G vs. T
**NDI**	I	31.1 ± 3.2	20.6 ± 4.5	10.9 ± 2.4	d = 2.6	d = 7.14	<0.001 *	<0.001 *	<0.001 *
	C	32.2 ± 2	29 ± 3.9	28.1 ± 5.1	d = 1.03	d = 1.05			
***p*-Value** **95% C.I.**		0.6 [−2.28, 0.08]	<0.001 * [−10.27, −6.52]	<0.001 * [−18.9, −15.4]					
**Pain intensity**	I	5 ± 1.5	1.4 ± 1.2	0.5 ± 1	d = 2.65	d = 3.53	<0.001 *	<0.001 *	<0.001 *
	C	5.6 ± 1	2.9 ± 0.9	3.2 ± 1.6	d = 2.8	d = 1.7			
***p*-Value** **95% C.I.**		0.04 [−1.16, −0.03]	<0.001 * [−1.07, −0.12]	<0.001 * [−3.29, −2.1]					

**Table 4 jcm-11-06028-t004:** The changes in posture control outcomes in experimental and control groups vs. time. SPENT = smooth pursuit neck torsion test; OSI = biodex balance test; HRA = head repositioning error in rotation right and left side; I = interventional group; C = control group; G = group; T = time; G vs. T = group vs. time; all values are expressed as means ± standard deviation; C.I. [] = 95% confidence interval; Cohen’s d value = d; * indicates statistically significant difference.

		Baseline	10-Weeks	1-Year Follow-up	Cohen’s d 10-Weeks vs. Baseline	Cohen’s d 1-Year vs. Baseline	*p*-Value		
							G	T	G vs. T
**HRA** **Right**	I	3.4 ± 1.4	2.1 ± 1.3	2 ± 1.5	d = 1.4	d = 1.3	<0.001 *	<0.001 *	<0.001 *
	C	4 ± 1.5	2.7 ± 1.1	3.2 ± 1.6	d = 0.9	d = 0.51			
***p*-value** **C.I.**	0.06 [−1.24, 0.04]		0.02 * [−1.13, −0.06]	<0.001 * [−1.89, −0.5]					
**HRA** **Left**	I	4.3 ± 1.4	2.6 ± 1.4	1.8 ± 1.1	d = 1.21	d = 1.98	<0.001 *	<0.001 *	<0.001 *
	C	3.7 ± 1.6	2.9 ± 1.6	2.8 ± 1.2	d = 0.5	d = 0.63			
***p*-value** **C.I.**	0.07 [−0.06, 1.26]		0.3 [−0.96, 0.36]	<0.001 * [−1.51, −0.48]					
**SPENT**	I	0.41 ± 0.17	0.28 ± 0.1	0.18 ± 0.09	d = 0.93	d = 1.6	<0.001 *	<0.001 *	<0.001 *
	C	0.34 ± 0.16	0.3 ± 0.06	0.29 ± 0.12	d = 0.09	d = 0.35			
***p*-value** **C.I.**	0.06 [−0.003, 0.14]		0.48 [−0.06, 0.02]	<0.001 * [−0.15, −0.06]					
**OSI**	I	0.62 ± 0.13	0.46 ± 0.1	0.41 ± 0.2	d = 1.37	d = 1.24	<0.001 *	<0.001 *	<0.001 *
	C	0.57 ± 0.11	0.52 ± 0.16	0.58 ± 0.19	d = 0.364	d = −0.06			
***p*-value** **C.I.**	0.06 [−0.003, 0.103]		0.047 * [−0.11, −0.0007]	<0.001 * [−0.25, −0.08]					

## Data Availability

The datasets analyzed in the current study are available from the corresponding author on reasonable request.

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
