# Peer review of "Reduction of Thoracic Hyper-Kyphosis Improves Short and Long Term Outcomes in Patients with Chronic Nonspecific Neck Pain: A Randomized Controlled Trial"

_jcm, 2022, doi:10.3390/jcm11206028_

Round 1

Reviewer 1 Report

Methods

ICT-ITL (max) 

 More clear description of what "good" correlation is between the formetric and gold standard device is needed.

Multimodal program

More description of the amount of time per session receiving therapy and the intensity of therapy is needed. Was the duration of therapy per session different between groups?

Data analysis

I am concerned about the description of the statistical methods. Please provide a description of the statistical test used for the main outcome measures. Only the tests used for descriptive statistics are included, and these would be inappropriate for the main outcome measures according to the design of the study.

Also, please describe what correction was made for multiple comparisons for the analysis of primary outcomes.

I would not consider the effect size derived from Cohen's formula to be an "r" value.

The procedures for imputing missing values needs to be clarified. How many missing values? What regressors were used? How strong was the effect of the regressions (R2)?

Results

The structure of the results and ordering of tables is inconsistent with the stated ordering of outcome measures. Also, I see little logic in the relative amount of results discussed for each outcome measure. Formatting is inconsistent. This makes the paper difficult to read and interpret the primary results of the paper. Please restructure. 

Figures

Please restructure figure order. The CONSORT flow diagram should be first.

Discussion and Conclusions

The caveat that the results only apply to patients with nonspecific neck pain who have thoracic kyphosis needs to be more clearly articulated throughout.

Table 1

Categorizing pain in 1-5 and >5 years is insufficient as pain catastrophization occurs ~2 years. Please provide greater granularity.

Table 2

I do not see a need to include an r value nor an SEM.

Reviewer 2 Report

Dear authors:  Thank you for the interesting study regarding treatment of hyper-kyphosis and the use of an orthotic as a treatment.  Overall the manuscript reads well although it comes across as fairly observational.  The paper would be improved with at least a hypothesis as to how the use of the orthotic may have contributed to the observed benefit.  Adults with a 'fixed' hyper-kyphosis would seem to not be amenable to physical maneuvers as opposed to perhaps functional deformities?  If the kyphosis changed significantly how to you hypothesize this occurred?  Is there anyway you could provide a hypothesis-based reasoning for this type of impact? 
